# On the production of ancient Egyptian blue: Multi-modal characterization and micron-scale luminescence mapping

Linda M. Seymour[1], Marco Nicola[2,3], Max I. Kessler[1], Claire L. Yost[1], Alessandro Bazzacco[4], Alessandro Marello[3], Enrico Ferraris[5], Roberto Gobetto[2], Admir Masic[1] *

1 Department of Civil and Environmental Engineering, Massachusetts Institute of Technology, Cambridge, MA, United States of America, 2 Dipartimento di Chimica, Università degli Studi di Torino, Torino, Italy, 3 Adamantio S.r.l, Torino, Italy, 4 Multispectral Imaging Freelance; Adjunct Professor in Multispectral Analysis, SUSCOR, Università degli Studi di Torino, Venaria Reale (TO), Italy, 5 Museo Egizio, Torino (TO), Italy

* masic@mit.edu

**Data Availability Statement:** All relevant data are within the paper and its Supporting Information files.

## Abstract

The ancient pigment Egyptian blue has long been studied for its historical significance; however, recent work has shown that its unique visible induced luminescent property can be used both to identify the pigment and to inspire new materials with this characteristic. In this study, a multi-modal characterization approach is used to explore variations in ancient production of Egyptian blue from shabti statuettes found in the village of Deir el-Medina in Egypt (Luxor, West Bank) dating back to the New Kingdom (18th-20th Dynasties; about 1550–1077 BCE). Using quantitative SEM-EDS analysis, we identify two possible production groups of the Egyptian blue and demonstrate the presence of multiple phases within samples using cluster analysis and ternary diagram representations. Using both macro-scale non-invasive (X-rays fluorescence and multi-spectral imaging) and micro-sampling (SEM-EDS and Raman confocal microspectroscopy) techniques, we correlate photoluminescence and chemical composition of the ancient samples. We introduce Raman spectroscopic imaging as a means to capture simultaneously visible-induced luminesce and crystal structure and utilize it to identify two classes of luminescing and non-luminescing silicate phases in the pigment that may be connected to production technologies. The results presented here provide a new framework through which Egyptian blue can be studied and inform the design of new materials based on its luminescent property.

## Introduction

Egyptian blue (EB) is a copper-based silicate material from antiquity and is credited as being the first synthetic pigment with the earliest known use dated to 3300–3200 BCE [1,2]. The primary component of EB is crystalline $CaCuSi_4O_{10}$, a compound analogous to the naturally occurring, rare mineral cuprorivaite [3–6]. The pigment became widely used during the early Egyptian dynasties (27th century BCE), and its steady production continued through the

**Funding:** This work is part of an ongoing project coordinated by E. F. and funded by the Fondazione Museo Egizio di Torino. The funders had no role in study design, data collection and analysis, decision to publish, or preparation of the manuscript.

**Competing interests:** The authors have declared that no competing interests exist.

Roman Empire until approximately the fifth century CE [7]. Its use gradually declined during the Early Middle Ages until its abandonment, although some instances of use are known beyond this period [7,8]. While still studied for its historical significance [7,8], the unique photoluminescence of EB has made it the subject of inspiration in emerging modern applications in fields such as biomedical imaging and security inks [9,10].

The first known procedure for EB production was documented by the ancient Roman scholar Vitruvius (VII.XI.1) [11]. EB can be produced artificially through a variety of methods, including solid state synthesis [12], solution combustion synthesis [13], hydrothermal synthesis [14], and melt flux synthesis, which is likely the most similar to the ancient method of production [12]. Generally, the pigment is made by mixing silica (*e.g.* silica gel or quartz powder) with calcium compounds (*e.g.* calcium carbonate, calcium hydroxide) and a copper source (*e.g.* copper oxide, bronze scrapings, malachite) in amounts approximating the ratio 4:1:1 for $SiO_2$:CaO:CuO [7]. Adding a flux, such as natron (predominately sodium carbonate) or plant ash (predominantly sodium and potassium carbonates), reduces the necessary heating temperature from almost 1000°C to around 900°C or even less [7]. The mixture is then heated for times varying from 10 to 100 hours [7].

The resulting EB pigment has a nano-sheet structure [9] and can be identified by its luminescent property [15], which is currently under investigation for the aforementioned applications. Specifically, the cuprorivaite absorbs a broad visible and near infrared light band (430–800 nm) and emits a narrow luminescent band in the infrared region (~910 nm) [16–18]. When the cuprorivaite phase is excited with visible light, three different transitions from the ground state occur: $^2B_{1g} \rightarrow {}^2B_{2g}$, $^2E_g$ and $^2A_{1g}$ [15,17,19,20]. When the ions return to their ground state, an emission spectrum centered at ~910 nm is released from the lowest energy electronic transition ($^2B_{2g} \rightarrow {}^2B_{1g}$), producing the characteristic luminescence [15]. The photoluminescence of EB can be visualized using visible induced luminescence (VIL) imaging technology, which captures the emission in the infrared range as an object is irradiated with visible light [19]. Using a modified digital camera, the spatial distribution of the pigment can be seen over the entire artifact. Pigment grains with dimensions on the order of 100 μm are detectable using VIL [21], enabling large scale applications, even if a majority of the pigment has degraded and the remnants are invisible to the human eye [22]. It is for this reason, among others, that the pigment is of interest for modern applications in a variety of industries.

In this work, a multi-modal characterization approach has been used to demonstrate the heterogeneous phase composition of ancient EB samples, correlate composition to possible production groups and map luminescing and non-luminescing silicate phases. The samples are from a collection of 54 shabti statuettes found in the village of Deir el-Medina in Egypt (Luxor, West Bank) dating back to the New Kingdom (18th-20th Dynasties; about 1550–1077 BCE) and curated by the Egyptian Museum in Turin, Italy [23]. The first non-invasive analysis utilized X-rays fluorescence (XRF) and multispectral imaging (visible reflected photography, near infrared (NIR) reflected photography and VIL) to macroscopically map the presence of the pigment in the shabti statuettes. EB from 10 statuettes was then micro-sampled and studied at higher spatial resolution using SEM-EDS and Raman microspectroscopy. The methodology presented here contextualizes the production of EB in antiquity by identifying compositional differences between pigment samples and correlating the luminescence to specific crystalline grains within a single pigment sample. The techniques described herein provide a means of characterizing EB and a starting point for future design work aimed at harnessing the photoluminescent property of the material.

## Results

XRF Spectroscopy (Table 1) was used for semi-quantitative elemental analysis of the pigments present on the shabti statuettes. Copper and calcium, two components of cuprorivaite, in addition to minor amounts of iron, potassium, and arsenic were detected in the areas visually identified as possibly containing EB (blue and black regions). The data suggest that EB was used as the blue pigment and, on some statuettes, it subsequently degraded, resulting in the black color observed [24]. The presence of EB was later confirmed using VIL (Figs 1 and S1).

VIL imaging (Figs 1 and S1) of the shabti statuettes confirmed and mapped the presence of EB. The pigment was found predominantly in the headdresses of the statuettes (*e.g.* Fig 1B) with additional EB found in hieroglyphs, necklaces, stripes or other decorative elements on the body (S1 Fig). To enhance the differences in the intensity of luminescence and to take into account the differences in the reflectance/absorption of NIR and visible light, a false-color approach was used [25]. After imaging, NIR false color (infrared false color, IRFC) and VIL false-color (VILFC) are blended using 50% opacity (S1 Fig). The composite result, represented in Table 1, is a range in color from black (low luminescence, low NIR reflectance) to red (high luminescence, high NIR reflectance). For the statuettes studied, we observed no direct correlation between the XRF data (Cu, Ca, Fe intensities) and the intensity of luminescence/ reflectance.

To understand the composition of the pigment distinct from the substrate to which it was applied, SEM-EDS analysis was performed on the 10 micro-samples. Fig 2 shows a representative sample of these datasets (Sample 2526, Fig 2A). From the EDS data, the presence of key elements in EB (Ca, Si, Cu) was explored in further detail. Quantified concentrations, as relative atomic percent, of each of these elements were plotted on ternary axes (Fig 2B) and

**Table 1. Multispectral imaging (presented as false color based on composite images in S1 Fig) and XRF results indicate the presence of luminescence and key elements associated with EB production.**

| Cat. Number | HUE Lab RGB | XRF (cps) | | | | |
|---|---|---|---|---|---|---|
| | | Cu | Ca | Fe | As | K |
| 2540 | 23; 11; -1 67; 52; 59 | 102 | 36 | 18 | 13 | 2 |
| 2764 | 29; 8; -2 77; 66; 73 | 542 | 135 | 52 | 25 | 8 |
| 2533 | 19; 9; 1 57; 45; 48 | 360 | 171 | 34 | 20 | 25 |
| 2601 | 21; 24; 18 76; 39; 32 | 352 | 136 | 60 | 23 | 8 |
| 2526 | 51; 35; 15 160; 95; 97 | 416 | 65 | 62 | 134 | 7 |
| 2777 | 31; 33; 13 105; 51; 56 | 670 | 93 | 38 | 13 | 5 |
| 2636 | 29; 40; 30 109; 38; 31 | 649 | 29 | 25 | 16 | 3 |
| 2530 | 14; 28; 13 63; 21; 25 | 550 | 79 | 32 | 12 | 5 |
| 2529 | 19; 26; 13 72; 32; 34 | 582 | 62 | 29 | 13 | 4 |
| 2538 | 25; 35; 15 95; 37; 44 | 688 | 118 | 34 | 20 | 7 |

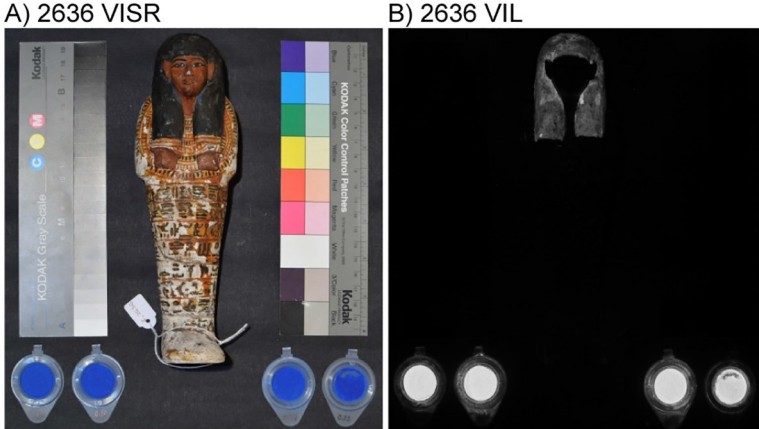

**Fig 1. Example of VIL imaging.** (A) Reflected visible light image of sample 2636. (B) Corresponding VIL image showing Egyptian blue is in the head dress. Egyptian blue standard is shown at the bottom of the images.

colored according to the density of points in the region. We can subsequently identify the dominant composition of the sample. Through this data-processing method, we observed that the centroid of the data is not consistent between samples (S2 Fig), nor is it where pure cuprorivaite would be expected (Ca:Si:Cu = 1:4:1). The quantified EDS data were fit with a Gaussian distribution to quantify the centroid and variance visible in the ternary plots (Fig 2B). The

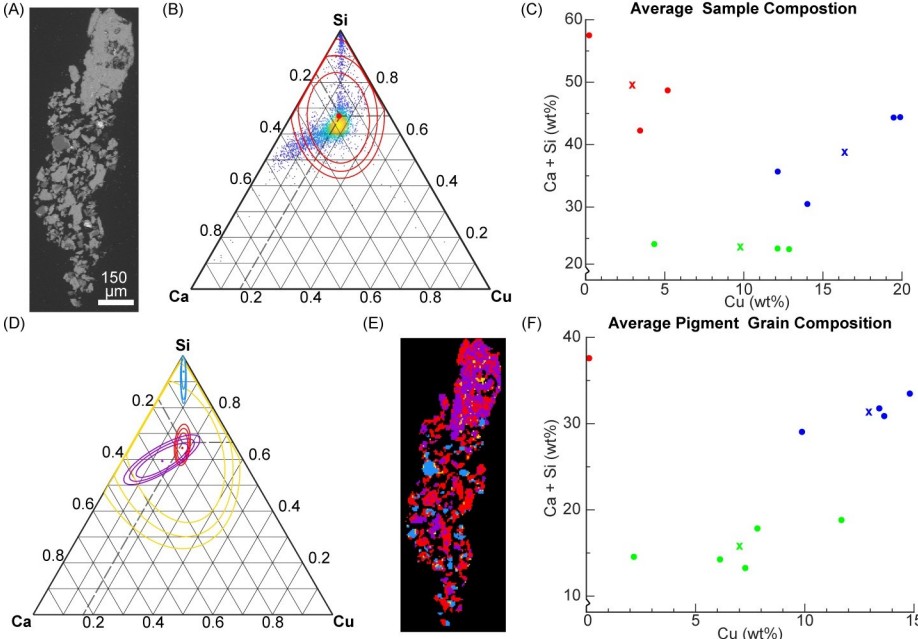

**Fig 2. EDS analysis of a representative sample, 2526.** (A) Back-scattered electron (BSE) image. (B) All EDS data is plotted on a ternary diagram where yellow represents the highest density of points. The centroid and 3 standard deviations of the data are plotted in red. (C) The centroid of all EDS samples colored according to k-means cluster groups. (D) EDS data replotted as data clusters assigned by the Gaussian mixture method. (E) The pixels of the EDS map are recolored according to the cluster to which they belong, corresponding to the color scheme in (D). (F) The centroid of each Gaussian mixture cluster corresponding to pigment grains for each shabti sample, colored according to the groups identified using k-means clustering. The composition of cuprorivaite is marked by dashed lines in (B) and (D). In (C) and (F) the centroid of each group is represented with an "x".

mean and three standard deviations were used to compare the EDS data of each sample. Using the quantified centroid of the elemental data, the samples were grouped using k-means clustering. Three groups were evident using this method and the centroids for each group showed variance in both the copper concentration and fraction of silicon and calcium present (Fig 2C). This clustering method, however, still included the composition of phases that are not EB such as remnant raw materials (*e.g.* quartz) and the ground layer (i.e. the layer on which pigment is applied). To refine the grouping of the pigment specifically, the EDS data for each sample were clustered using Gaussian mixture modelling in order to distinguish the pigment grains from other sample components. The results were once again plotted on ternary axes (Fig 2D). The clustering results of the sample data were then visualized on the EDS map (Fig 2E) by recoloring each pixel to correspond to the cluster to which it had the highest probability of belonging. The intra-sample clustering (i.e. clustering the data points of a single sample) groups pixels that have similar elemental composition, highlighting subtle differences in chemistry. This further refinement of the data allowed the samples to be grouped based on pigment composition, identified as the EDS data Gaussian cluster for which the centroid had the highest concentration of copper (e.g. the red cluster in Fig 2D). The samples were grouped using k-means clustering of the centroids corresponding to the presumed EB pigment grains (Fig 2F). Thus, Fig 2F represents the average composition of isolated EB pigment grains in each shabti sample and the correspond group to which the sample belongs. Two groups and one outlier were apparent using this method. The outlier, sample 2530, was confirmed to be a sample of ground layer, lacking any substantial pigment grains. The two primary groups are similar in composition to two groups from the bulk EDS data (S1 and S2 Tables); however, the centroids show lower fractions of calcium, likely because the influence of non-copper containing phases such as the ground layer have been removed.

VIL imaging provides an overview of EB distribution on an artifact; however, to develop the technology for modern applications, higher resolution mapping and correlation with chemical features, such as those identified using SEM-EDS, is highly desirable. Raman spectroscopy, as schematically described in Fig 3A, was used to probe the phases present in the pigments because of the two distinct groups seen in the latter clustering methodology. Raman spectroscopy is a technique wherein a monochromatic laser light ($\lambda_0$), typically in the visible spectrum, is directed onto a sample and interacts with the electron cloud. Most of the incident light is scattered elastically (Rayleigh scattering, $\lambda = \lambda_0$); however, in some cases, the incident light is modulated by vibrational modes of the molecule or crystal resulting in inelastic (Raman) scattering ($\lambda = \lambda_0 \pm \Delta$). In the case of cuprorivaite, however, the incident visible light is absorbed and re-emitted in the infrared ($\lambda \sim 910$ nm) in addition to the Raman scattering. The Raman spectrometer can capture each of these instances, Raman scattering and photoluminescence, thus allowing unprecedented high-resolution characterization of EB samples. By scanning each pigment sample twice, once to capture low wave number shifts corresponding to Raman scattering and once to capture the VIL emission, we can achieve micron-scale mapping of the pigment. Whereas standard VIL imaging provides a macroscopic distribution of luminescent pigment on an object, this spectroscopic technique distinguishes the distribution of luminescence within individual grains and correlates it to specific mineral phases.

The composition of pigment grains (Fig 3D, 3F and 3H) was mapped at a spatial resolution of 2–4 μm (Fig 3E, 3G and 3I). Two Raman spectra of the pigment grains were identified (Fig 3B), indicating that the pigments contain two different crystalline components. Only one of these spectra, identifiable by a peak at 981 cm$^{-1}$, correlated directly with the characteristic luminescence (Fig 3C) in the pigment. Both pigment phases that were identified were consistent among the samples scanned.

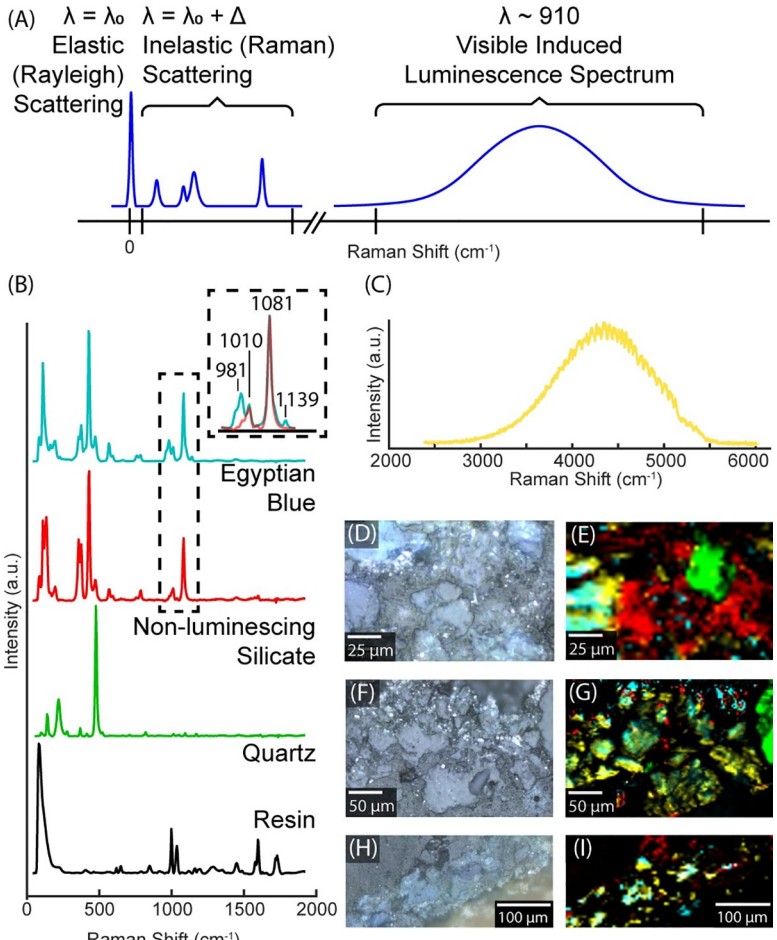

**Fig 3. Spectroscopic mapping of phases and luminescence.** (A) Schematic of the Raman and photoluminescence spectra acquired. (B) Component spectra identified in the Raman maps. (C) Spectrum of characteristic luminescence spectrum identified. (D) Optical image of sample 2526. (E) Phase and luminescence map of the area in (D). (F) Optical image of sample 2636. (G) Phase and luminescence map of the area in (F). (H) Optical image of sample 2601. (I) Phase and luminescence map of the area in (H). Phase and luminescence maps are colored according the Raman spectra in (B) and luminescence spectrum in (C).

To analyze further the Raman spectra for the pigment phases identified, Gaussian curves were fit to the Si-O bending (~500 cm$^{-1}$) (S3 Fig) and stretching (~1000 cm$^{-1}$) (Figs 4 and S4) bands. The relative area of these two regions ($A_{500}/A_{1000}$) can be used to estimate the polymerization index ($I_p$) of glassy silicate networks [26]. Both the luminescing ($I_p = 1.10$) and non-luminescing phases ($I_p = 1.39$) have polymerization indices associated with sodium and calcium rich fluxes which is to be expected for ancient EB production [26]. Further analysis of the Si-O stretching region (Fig 4 insets) shows an increase in the relative amount $Q_2$ linkages for the luminescing phase ($A_{Q2}/A_{Q1-4} = 0.32$) compared to the non-luminescing phase ($A_{Q2}/A_{Q1-4} = 0.06$).

Luminescence mapping of the pigment samples over large areas at high resolution revealed the variation in luminescence intensity within pigment grains. A sample from each identified pigment group in Fig 2F was selected for Fig 5. Sample 2526 (Fig 5A–5D) exhibits pigment grains with copper rich, calcium rich and silicon rich regions, as evidenced by EDS mapping (Fig 5B). These regions cluster together revealing a heterogeneous pigment sample (Fig 5C).

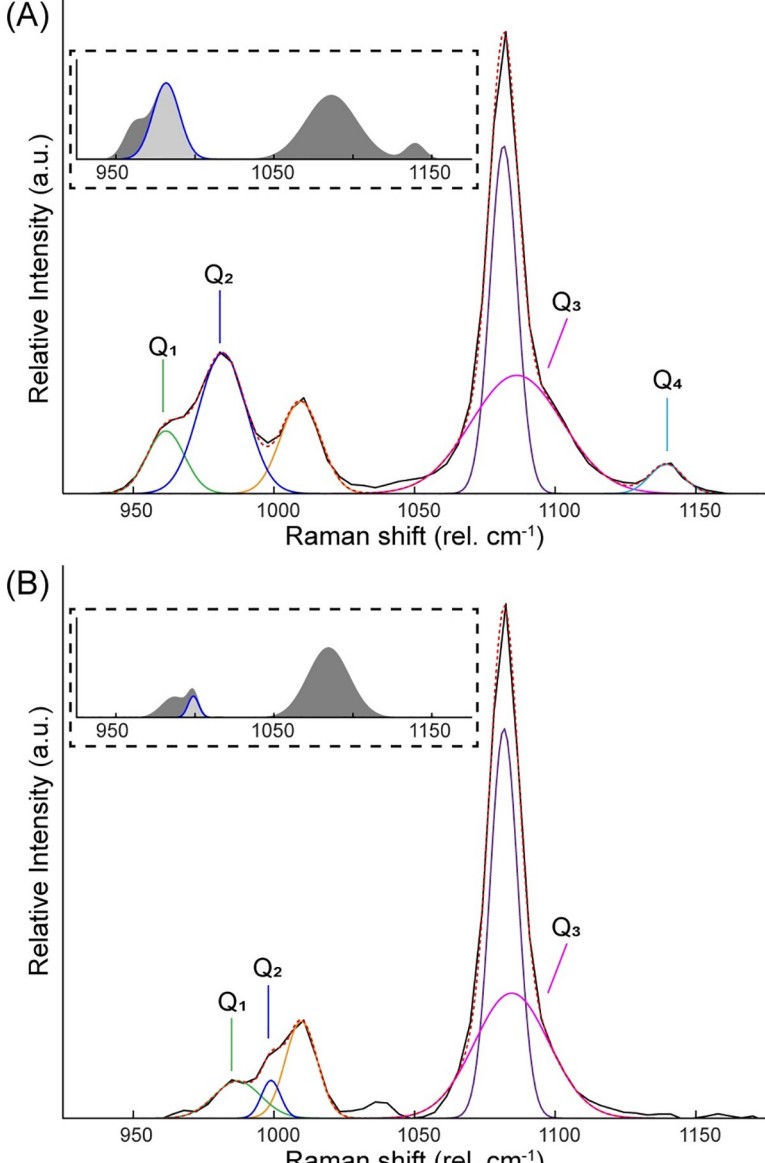

**Fig 4. Analysis of Raman spectra for identified pigment phases.** The Si-O stretching region of the Raman spectra for (A) Egyptian blue and (B) the silicate glass phase is analyzed by fitting Gaussian curves to the original spectra (black). The total fit, including contributions from sulfate and carbonate vibrations (1010 and 1081 cm-1 respectively), is indicated by the red dotted line. The insets show the total area associated with Si-O vibrations and the relative contribution of the $Q_2$ band.

This heterogeneity can also be seen through the luminescence mapping (Fig 5D) wherein it is clear that a single pigment grain can have both high-intensity luminescence and little to no luminescence occurring. On the contrary, sample 2601 (Fig 5E–5H) shows less intra-grain variation. In the EDS map, grains rich in silicon and the calcium rich ground layer are distinct from the pigment grains. Clustering of this data (Fig 5G) confirms that the pigment grains are relatively homogeneous in composition, which is further exemplified by luminescence (Fig 5H) that is consistent within grains, though the intensity varies from grain to grain.

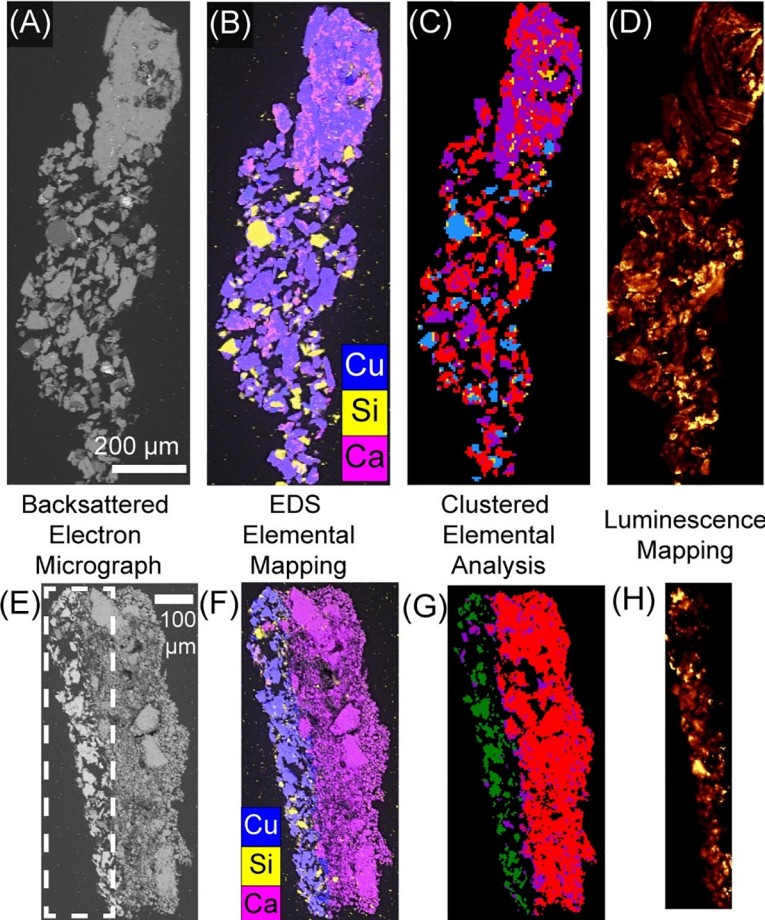

**Fig 5. Correlative characterization of pigment samples from the two EDS groups.** (A) EDS map of sample 2526 showing the pigment grains have copper rich, calcium rich and silicon rich regions. (B) Clustered data from (A) (as in Fig 1E) shows that these regions cluster together. (C) Luminescence map acquired reveals that intensity varies within pigment grains. D) EDS map of sample 2601 shows more homogeneous pigment grains with some silicon rich regions and calcium rich binder. E) Clustering of the EDS data in (D) confirms the homogeneity of the pigment grains. F) Luminescence map acquired of the region indicated by the dashed line in (E) shows that intensity varies between grains but less so within the pigment grains.

## Discussion

The presence of two pigment groups of different elemental compositions, luminescing and non-luminescing phases suggests that EB can occur in a variety of forms. Gaussian mixture modeling was employed to observe subtle differences in composition within each sample; however, we note that grains that appeared similar in EDS in some cases had distinct Raman spectra. Furthermore, we observe the same characteristic luminescence at the local scale despite the varied elemental compositions in the pigments identified using EDS. This indicates that the characteristic luminescent property used to identify EB relies on the crystalline structure that forms during its production.

Comparing the network of $SiO_4$ tetrahedra in the identified Raman phases, the relative increase in $Q_2$ linkages associated with the luminescing phase is consistent with the crystal structure of cuprorivaite wherein the $Cu^{2+}$ integrates into the sheet-like silicate network [16,27], replacing an Si-O-Si linkage. The relative intensity of the $Q_3$ peak in the non-luminescing phase and high index of polymerization suggest that the phase also has a sheet-like

structure. Furthermore, the correlation between the $Q_2$ linkages and the luminescence supports the underlying crystal structure as responsible for the phenomenon. The peak fitting presented here (Figs 4, S3 and S4) shows that while major features of the Raman spectra of both the luminescing and non-luminescing phases are similar, the subtle differences seen in the $SiO_4$ stretching regions can be used to definitively distinguish between EB (i.e. luminescing, cuprorivaite-based pigment) and the surrounding silicate material. The agreement in the $SiO_4$ bending regions of the spectra further show the similarities in structure between these two phases. We further observe that the higher index of polymerization in the non-luminescing phase could explain the shift in the $Q_1$ peak position, increasing the energy of the vibration.

While the combined Raman and EDS results support that the crystalline structure of cuprorivaite is responsible for the luminescence, the exact phenomenon is still unclear. The most commonly accepted explanation is based on the position of the $Cu^{2+}$ ion, sitting in square-planar coordination with surrounding $SiO_4$ tetrahedra. It is generally agreed that the interaction with the $SiO_4$ tetrahedra splits the electronic energy levels of the $Cu^{2+}$, causing a band gap that corresponds with the energy of the photoluminescence [16]. We observe with the combined Raman and luminescence mapping that the cuprorivaite grains are solely responsible for the luminescence in the pigment, despite the presence of another blue, sheet-silicate phase.

The two pigment groups could point towards different production methods that result in the formation of EB. In sample 2636, for example, which is less copper-rich according to the EDS data, most of the grains luminesced, at least partially, in the Raman map and showed consistent Raman spectra. Conversely, sample 2526, contained entire pigment grains that did not luminesce with Raman mapping, though there were distinct areas of luminescence present Most samples had remnants of silicon rich components (identified as quartz in those samples that were mapped with Raman), however remnants of other raw materials are not as easily identified using the techniques employed herein. Previous research has showed that repeated calcining and purification processes, such as dissolving excess CuO and $CaCO_3$ in HCl, can refine the purity of EB, resulting in better luminescent properties [28]. It is further evident from both elemental and luminescence mapping (Fig 5A–5C) that individual pigment grains are not purely cuprorivaite. The luminescing component of EB appears, in some cases, to be contained within a matrix exhibiting higher concentrations of silicon and calcium. As the Raman spectra of the non-luminescing grains is remarkably similar to the luminescing ones, this could be a semi-crystalline or crystalline precursor to cuprorivaite in the pigment (i.e. the result of insufficient purification or other production parameters such as cooling rate) or the result of improper proportions of raw materials to form pure cuprorivaite.

While it is evident from the Raman and luminescence mapping (Fig 3) that the crystallinity of the pigment grains plays a significant role in the pigment's luminescent property, areas of similar elemental composition may have different luminescence intensities (Fig 5). Thus, luminescent intensity cannot be directly correlated to a specific elemental composition using the methods employed herein. When trying to directly correlate luminescence to elemental composition, complications arise. Most notably, the strength of the luminescence signal coupled with the penetration depth of the laser could result in detection beyond the surface of the sample. This is evident in Fig 3 wherein areas identified as embedding medium (resin) in Raman still show luminescence. Still, using the combined EDS and Raman data sets holistically can provide complementary information as demonstrated here.

## Conclusion

High-resolution chemical characterization provides unique insights into the structure and composition of the ancient pigment. The multi-modal approach presented herein revealed the

variation in EB composition across 10 micro-samples taken from shabti statuettes of similar period and origin. Ternary diagram representations of quantitative EDS data show the varied composition both between samples and within individual samples. These differences can be related to different production parameters and, possibly, to different production workshops. Using a Raman spectrometer, we have shown that the visible induced luminescence phenomenon can be mapped and correlated to specific crystalline grains of the pigment. Interestingly, the luminescence does not correlate to a specific elemental composition detectable by the methods employed herein and instead relies on the crystallinity of the pigment itself. Further analysis of a wider sample set is required to generalize the conclusions presented here; however, we note that the combination of multi-modal techniques presented provide a framework to study the ancient pigment and modern materials inspired by it. The results suggest that different production techniques impact the luminescence of EB and subsequently inform future implementation of the material in various applications.

## Materials and methods

### Sample curation

All analysis carried out on the 10 wooden Ushabty which are the subject of the research, have been approved by Museo Egizio and have been authorized by the Soprintendenza Archeologia Belle Arti e Paesaggio per la città Metropolitana di Torino with document Prot.n. 17642 dated 16th of November 2017.

### Multispectral imaging

A modified digital camera (Nikon D3200, sensor CMOS DX, 24.7) with the inner IR and UV filter removed was used to acquire VIL and reflected NIR images. The camera has a range of about 1000 nm. To remove visible light (VIS) from the recorded images, an external B+W IR pas 830 filter was used. For VIL imaging a LED lamp (YONGNUO YN300) with low emission in the infrared (IR) range was used. For reflected IR imaging two IFF Q 1000 with 1000W halogen lamps (3400°K) were used.

### X-Ray fluorescence

XRF data were acquired using a portable XRF spectrophotometer (Assing LITHOS 3000), with monochromatic excitation energy (molybdenum Kα, voltage 25 kV, 0.3 mA) and a Si-PIN detector Peltier cooled. The measurement area was approximately 50 mm$^2$ and the distance between the tool and the surface was kept at 10 mm (standard distance). The accumulation time was set to 120 seconds.

### Scanning Electron Microscopy and Energy-Dispersive X-Ray Spectroscopy (SEM-EDS)

SEM-EDS data were collected on resin-embedded samples using a TESCAN RISE scanning electron microscope. All samples were imaged in low vacuum (30 Pa) with an acceleration voltage of 20 keV (air type: N2). Acquisition time for EDS data was 12 hours. EDS data were quantified using Bruker Esprit 2.1 Software with 2x2 data binning and Linemarker PB-ZAF correction. SEM-EDS data were further processed using custom MATLAB (R2019a) scripts. Gaussian mixture modelling using the MATLAB Statistics and Machine Learning Toolbox was used to cluster the data (as atomic percent of elements present) within each scan. For each model, 10 replicates were generated. Model performance was monitored using the negative log likelihood with the final model selected as having the best performance. Over-

fitting was avoided by selecting the least number of distributions required to improve the performance metric.

### Raman spectroscopy and luminescence mapping

Raman spectroscopy was performed with a WiTec Alpha 300R confocal Raman microscope. Samples were imaged using a Zeiss Epiplan-Neofluar 50x long-distance objective lens (NA 0.55). For each image, two scans were consecutively run without disrupting the sample. For phase identification a WiTec 75 mW Nd:YAG 532 nm laser at 1/3 power and 600 g/mm spectrometer grating was used. At each point in the scan areas of Fig 3, a single spectrum consisting of 3–5 averaged accumulations was obtained. Integration time varied from 1.78 to 5.10 seconds to improve peak to background ratio.100 single spectra were collected and averaged for a modern EB sample (Kremer Pigmente, New York) with an integration of 0.5 seconds per spectrum. For VIL mapping a Research Electro-Optics 35 mW helium-neon 633 nm laser at full power and 300 g/mm spectrometer grating was used. Due to the intensity of the luminescence, a single accumulation at each point in the scan area was used and accumulation time varied from 0.2 to 0.53 seconds. Lateral resolution of the scans varied from 2.5 to 5.0 μm. In Fig 5, the lateral resolution of VIL mapping for both samples in is 1.5 μm. The data were processed using WiTec's Project 5 software. The software's built in k-means clustering function was used to identify component spectra of each scan.

## Supporting information

**S1 Fig.** Visible (top), VIL (middle) and VILFC+IRFC (bottom) Imaging of the 10 statuettes from this study.
(TIF)

**S2 Fig. Ternary diagrams for the 10 samples analyzed by EDS demonstrating the shift of the centroid between samples.** Identities of the samples are (A) 2526; (B) 2529; (C) 2530; (D) 2764; (E) 2777; (F) 2533; (G) 2540; (H) 2601; (I) 2538; (J) 2636.
(TIF)

**S3 Fig. Gaussian fitting of the $SiO_4$ bending region of Raman spectra.** (A) Luminescing phase; (B) non-luminescing phase; (C) modern pigment.
(TIF)

**S4 Fig. Gaussian fitting of the $SiO_4$ stretching region of the Raman spectrum for modern Egyptian Blue.** The relative area of the $Q_2$ band (inset) is 0.199 and the approximated index of polymerization is 0.516.
(TIF)

**S5 Fig. Enlarged ternary diagram clarifying Fig 2D.**
(TIF)

**S6 Fig.**
(TIF)

**S1 Table. Sample grouping corresponding to Fig 2C.**
(DOCX)

**S2 Table. Sample grouping corresponding to Fig 2F.**
(DOCX)

## Acknowledgments

This work is part of an ongoing project coordinated by the curator of Museo Egizio E. Ferraris and led by A. G. de Marco. In compliance with the Italian regulations in place in the Conservation field, all necessary permits were obtained for the described study.

## Author Contributions

**Conceptualization:** Admir Masic.

**Data curation:** Linda M. Seymour, Marco Nicola, Max I. Kessler, Claire L. Yost, Alessandro Bazzacco, Alessandro Marello, Enrico Ferraris.

**Formal analysis:** Linda M. Seymour, Marco Nicola, Max I. Kessler, Claire L. Yost, Alessandro Bazzacco, Alessandro Marello.

**Funding acquisition:** Enrico Ferraris.

**Investigation:** Max I. Kessler, Claire L. Yost, Alessandro Bazzacco, Alessandro Marello, Roberto Gobetto.

**Methodology:** Linda M. Seymour, Marco Nicola, Max I. Kessler, Claire L. Yost, Alessandro Bazzacco, Alessandro Marello.

**Project administration:** Enrico Ferraris, Roberto Gobetto, Admir Masic.

**Resources:** Enrico Ferraris, Admir Masic.

**Supervision:** Roberto Gobetto, Admir Masic.

**Validation:** Marco Nicola, Enrico Ferraris.

**Visualization:** Linda M. Seymour, Marco Nicola.

**Writing – original draft:** Linda M. Seymour, Marco Nicola, Admir Masic.

**Writing – review & editing:** Admir Masic.

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
