## [Decision Letter · Decision Letter 0]

6 Oct 2020

PONE-D-20-25163

On the production of ancient Egyptian blue: multi-modal characterization and micron-scale luminescence mapping

PLOS ONE

Dear Dr. Masic,

Thank you for submitting your manuscript to PLOS ONE. After careful consideration, we feel that the manuscript has merit but could be improved to fully meet PLOS ONE’s publication criteria. Therefore, we invite you to submit a revised version of the manuscript that addresses the points raised during the review process.

In particular, please address the comments raised by the Reviewer #2 and make corresponding changes in the manuscript.

We look forward to receiving your revised manuscript.

Kind regards,

Oksana Ostroverkhova

Academic Editor

PLOS ONE

Journal Requirements:

Reviewers' comments:

Reviewer's Responses to Questions

**Comments to the Author**

1. Is the manuscript technically sound, and do the data support the conclusions?

Reviewer #1: Yes

Reviewer #2: Yes

2. Has the statistical analysis been performed appropriately and rigorously? 

Reviewer #1: Yes

Reviewer #2: Yes

3. Have the authors made all data underlying the findings in their manuscript fully available?

Reviewer #1: Yes

Reviewer #2: Yes

4. Is the manuscript presented in an intelligible fashion and written in standard English?

Reviewer #1: Yes

Reviewer #2: Yes

5. Review Comments to the Author

Reviewer #1: This short paper was enjoyable to read, exciting data, and relevant/refreshing to what I had considered a mature research topic. The micro-spectroscopy of archaeological Egyptian blue (EB) samples from the Egyptian Museum in Torino were studied. The paper's critical insight is that a blue Raman laser (532nm) can both induce the Raman effect and photoluminescence of cuprorivaite in a single experiment, thus acting as a probe to examine grain heterogeneity. Their findings show that, in some cases, regions of an individual EB pigment grain may not luminesce when excited with visible light. This finding could be due to these regions not having formed into a structured crystal or, possibly, due to decomposition due to furnace temperature fluctuations (or excess flux in that area, etc.). I believe more correlative spectroscopy could answer this, and it is a suggestion that the authors look into synchrotron-based micro-diffraction as another way to probe these structures.

These findings lay the ground for experimental replication work into how EB forms and how crystallization occurs out of a flux melt. For the archaeological scientist, such work will help define "workshop practices" of these ancient craftsmen.

Reviewer #2: The paper is a highly challenging analysis of Egyptian artefacts using combined EDX, Raman and micro-luminescence techniques.

The skills of the scientists lead to consistent insights on the nature of luminescence and chemical composition and differences in the pigment layer in the shabti statuette series.

Some clarifications should be implemented in the text:

1. table 1 is almost undescribed, in terms of caption, legend and also underlying method (i.e. IRC is not declared). According to the text, the table could also report samples where traces of heteroelements (K, As, Fe) were found.

2. lines 113-122: false color images are explained but are not shown neither in the text nor in the SI.

3. in figure 2, lines showing the expected cuprorivaite composition could be added (as done in the plots in the SI).

4. in figure 2, the "x" of the centroids are poorly visible, and would be useful to make them in the colour referred to the circles they are related with.

5. Maybe it is unclear to me, but in the ternary diagram if figure 2d there are 4 gaussian fit distribution but in plot 2f only 3 colored series are shown. Moreover, red circles in 2f are overlapped, I guess...

6. line 165 on: differences between Raman and photoluminescence are well explained, but in the text (maybe since they are collected with the same equipment) are often used in misleading way. Each time the collected response is the 910 nm response, the technique does not deal with Raman effect but (consistently with the title of the paper) with micron-scale luminescence. The text should be arranged using these guidelines.

7. line 190: I suppose that the referred resolution is the spatial resolution of 2-4 microns. The notation "per pixel" seems to me uncorrect.

6. PLOS authors have the option to publish the peer review history of their article (what does this mean?). If published, this will include your full peer review and any attached files.

Reviewer #1: No

Reviewer #2: No

---

## [Author Response · Author response to Decision Letter 0]

2 Nov 2020

We have addressed all the comments and suggestions raised by reviewers and ensured that the revised manuscript complies with the format requirements. 

Answers to Reviewers’ Comments 

Reviewer #1: 

This short paper was enjoyable to read, exciting data, and relevant/refreshing to what I had considered a mature research topic. The micro-spectroscopy of archaeological Egyptian blue (EB) samples from the Egyptian Museum in Torino were studied. The paper's critical insight is that a blue Raman laser (532nm) can both induce the Raman effect and photoluminescence of cuprorivaite in a single experiment, thus acting as a probe to examine grain heterogeneity. Their findings show that, in some cases, regions of an individual EB pigment grain may not luminesce when excited with visible light. This finding could be due to these regions not having formed into a structured crystal or, possibly, due to decomposition due to furnace temperature fluctuations (or excess flux in that area, etc.). I believe more correlative spectroscopy could answer this, and it is a suggestion that the authors look into synchrotron-based micro-diffraction as another way to probe these structures.

These findings lay the ground for experimental replication work into how EB forms and how crystallization occurs out of a flux melt. For the archaeological scientist, such work will help define "workshop practices" of these ancient craftsmen.

 We thank the reviewer for their comments and will continue to explore techniques such as micro-diffraction to further expand this work. 

Reviewer #2: 

The paper is a highly challenging analysis of Egyptian artefacts using combined EDX, Raman and micro-luminescence techniques.

The skills of the scientists lead to consistent insights on the nature of luminescence and chemical composition and differences in the pigment layer in the shabti statuette series.

We thank the reviewer for their comments and have updated the manuscript based on the clarifications mentioned below.

Some clarifications should be implemented in the text:

1. table 1 is almost undescribed, in terms of caption, legend and also underlying method (i.e. IRC is not declared). According to the text, the table could also report samples where traces of heteroelements (K, As, Fe) were found.

We have updated Table 1 and expanded our explanation of the presented data. 

2. lines 113-122: false color images are explained but are not shown neither in the text nor in the SI.

Thank you for this observation. We have included further supporting information with the full false color images as well as additional text describing the data.

3. in figure 2, lines showing the expected cuprorivaite composition could be added (as done in the plots in the SI).

We appreciate this suggestion and have updated the plots accordingly.

4. in figure 2, the "x" of the centroids are poorly visible, and would be useful to make them in the colour referred to the circles they are related with.

We thank the reviewer for this suggestion and have updated the plots to be easier to read.

5. Maybe it is unclear to me, but in the ternary diagram if figure 2d there are 4 gaussian fit distribution but in plot 2f only 3 colored series are shown. Moreover, red circles in 2f are overlapped, I guess...

We have added Tables S1 and S2 in the supplementary materials to clarify that the in 2c and 2f points represent individual samples whereas the Gaussian clustering is for phases within a single sample. The red grouping in Fig 2d has a very low standard deviation so the lines appear overlapping due to the line width, but are distinct ellipses. We have added a supplemental figure of an enlarged figure 2d to clarify this.

6. line 165 on: differences between Raman and photoluminescence are well explained, but in the text (maybe since they are collected with the same equipment) are often used in misleading way. Each time the collected response is the 910 nm response, the technique does not deal with Raman effect but (consistently with the title of the paper) with micron-scale luminescence. The text should be arranged using these guidelines.

We appreciate this observation and agree that we should distinguish between the Raman scattering and observed luminescence. We have updated the text accordingly.

7. line 190: I suppose that the referred resolution is the spatial resolution of 2-4 microns. The notation "per pixel" seems to me uncorrect.

We thank the reviewer for this comment and have reworded the sentence accordingly.

---

## [Editor Report · Decision Letter 1]

5 Nov 2020

On the production of ancient Egyptian blue: multi-modal characterization and micron-scale luminescence mapping

PONE-D-20-25163R1

Dear Dr. Masic,

We’re pleased to inform you that your manuscript has been judged scientifically suitable for publication and will be formally accepted for publication once it meets all outstanding technical requirements.

Kind regards,

Oksana Ostroverkhova

Academic Editor

PLOS ONE
---

## [Editor Report · Acceptance letter]

12 Nov 2020

PONE-D-20-25163R1 

On the production of ancient Egyptian blue: multi-modal characterization and micron-scale luminescence mapping 

Dear Dr. Masic:

I'm pleased to inform you that your manuscript has been deemed suitable for publication in PLOS ONE. Congratulations! Your manuscript is now with our production department. 

Kind regards, 

on behalf of

Prof. Oksana Ostroverkhova 

Academic Editor

PLOS ONE